# Insect Collections as an Untapped Source of Bioactive Compounds—Fireflies (Coleoptera: Lampyridae) and Cardiotonic Steroids as a Proof of Concept

**DOI:** 10.3390/insects12080689

**Published:** 2021-07-31

**Authors:** Andreas Berger, Georg Petschenka, Thomas Degenkolb, Michael Geisthardt, Andreas Vilcinskas

**Affiliations:** 1Institute for Insect Biotechnology, Justus Liebig University Giessen, Heinrich-Buff-Ring 26–32, 35392 Giessen, Germany; BAndreas90@web.de (A.B.); Thomas.Degenkolb@ernaehrung.uni-giessen.de (T.D.); 2Department of Applied Entomology, Institute of Phytomedicine, Faculty of Agricultural Sciences, University of Hohenheim, Otto-Sander-Strasse 5, 70599 Stuttgart, Germany; 3Rathenaustraße 9b, 61184 Karben, Germany; michael.geisthardt@t-online.de; 4Department of Bioresources, Fraunhofer Institute for Molecular Biology and Applied Ecology, Ohlebergs-weg 12, 35392 Giessen, Germany

**Keywords:** Lampyridae, fireflies, lucibufagins, bufadienolides, cardiac glycosides, natural history collections, natural products

## Abstract

**Simple Summary:**

Natural history museums around the world possess extensive collections of dried insects (e.g., beetles, butterflies or flies). Traditionally, museum specimens were mostly studied anatomically for systematic (i.e., how different insects are related) and taxonomic purposes (e.g., description of new species). During the last decades, it has become a common practice to study insect systematics based on DNA extracted from dried museum specimens. Being a highly sought-after prey, many insects evolved powerful toxins to ward off predators. One example is seen in fireflies, a beetle family famous for their ability to emit light. A few species of mainly North American fireflies, however, are known to produce toxins that are called lucibufagins. In this study, we tested if lucibufagins also occur in European species of fireflies. Instead of sampling in the field, we chemically analyzed firefly specimens from museum collections by using a method that preserves the valuable museum specimens. In total, we found lucibufagins in 21 species of fireflies, including specimens that were older than 100 years. Our study emphasizes that insect collections provide a valuable archive of chemical information that can be used for the discovery of novel pharmacologically interesting compounds as well as for addressing ecological questions without destroying valuable specimens.

**Abstract:**

Natural history collections provide an invaluable basis for systematics, ecology, and conservation. Besides being an important source of DNA, museum specimens may also contain a plethora of natural products. Especially, dried insect collections represent a global repository with billions of inventoried vouchers. Due to their vast diversity, insects possess a great variety of defensive compounds, which they either produce autogenously or derive from the environment. Here, we present a case study on fireflies (Coleoptera: Lampyridae), which produce bufadienolides as a defense against predators. These toxins belong to the cardiotonic steroids, which are used for the treatment of cardiac diseases and specifically inhibit the animal enzyme Na^+^/K^+^-ATPase. Bufadienolides have been reported from only seven out of approximately 2000 described firefly species. Using a non-destructive approach, we screened 72 dry coleopteran specimens for bufadienolides using HPLC-DAD and HPLC-MS. We found bufadienolides including five novel compounds in 21 species of the subfamily Lampyrinae. The absence of bufadienolides in the phylogenetically related net-winged beetles (Lycidae) and the lampyrid subfamilies Luciolinae and Lamprohizinae indicates a phylogenetic pattern of bufadienolide synthesis. Our results emphasize the value of natural history collections as an archive of chemical information for ecological and evolutionary basic research and as an untapped source for novel bioactive compounds.

## 1. Introduction

Natural history collections are of great value, not only for taxonomic, systematic, and morphological research, but also for the documentation of changes in biodiversity [1], including the period of accelerated anthropogenic influences on habitats and climate [2], the study of evolutionary processes [3], and as a resource for innovative education [4]. Although museum specimens are likely to contain a great variety of chemical compounds with various ecological functions, only few studies analyzed chemical substances derived from museum specimens [5,6,7]. Here, we corroborate that natural history collections can serve as an untapped repository for natural products that can be obtained from specimens that are more than 100 years old. As a case study, we investigated European firefly species (Lampyridae) for the occurrence of cardiotonic steroids. 

Fireflies (Lampyridae) comprise roughly 2000 species in 83 genera, showing the highest biodiversity in the Neotropics and the oriental region [8]. While being famous for the emission of light pulses during courtship behavior, fireflies also produce lucibufagins, a group of highly toxic cardiotonic steroids. Lucibufagins in fireflies were first described by Eisner et al. from the north North-American species *Photinus marginellus* (LeConte 1852) [9]. In addition, these compounds were found in *Photinus ignitus* (Fall 1927) [10], *Photinus pyralis* (Linnaeus 1767) [11,12], *Photuris versicolor* (Fabricius 1798) [13,14], *Lucidota atra* (Olivier 1790) [15], and recently *Ellychnia corrusca* (Linnaeus 1767) [16] and *Diaphanes lampyroides* (Olivier 1891) [17]. While almost all records of lucibufagins in fireflies were made for North-American species, Tyler et al. were the first who reported the occurrence of these compounds in the Eurasian species *Lampyris noctiluca* (Linnaeus 1767) [18].

Lucibufagins belong to the bufadienolides, which, together with the cardenolides, form a group of steroidal compounds called cardiotonic steroids or cardiac glycosides, a name indicating their wide use for the treatment of congestive heart failure. Cardenolides possess a five-membered monounsaturated lactone ring in the β position at C_17_, whereas the bufadienolides carry a six-membered doubly unsaturated lactone ring at the same position [19]. Both groups are specific inhibitors of the ubiquitous animal enzyme Na^+^/K^+^-ATPase [20]. While cardenolides (especially digoxin and digitoxin) had an important role for the treatment of congestive heart failure and supraventricular arrhythmias [20], the medical use of bufadienolides is much more restricted. Nevertheless, bufadienolides are components of the important traditional Chinese medicine ChanSu that is derived from toad skin secretions [21]. Moreover, bufadienolides possess a wide range of biological activities, among which their antitumor activity is considered as being the most interesting one [22].

While bufadienolides are known to occur in six plant families (Asparagaceae, Crassulaceae, Iridaceae, Melianthaceae, Ranunculaceae, and Santalaceae), their occurrence in animals is rare [22]. Besides fireflies, bufadienolides are only found in toads (Bufonidae) and snakes of the genus *Rhabdophis* (Colubridae) that sequester bufadienolides from toads [23] and firefly larvae [17]. Moreover, the milkweed bug *Spilostethus pandurus* (Heteroptera: Lygaeinae) sequesters bufadienolides from squill (*Urginea maritima*, Asparagaceae) [7]. Within fireflies, the toxins were only found in species of the subfamily Lampyrinae and in the genus *Photuris* (Photurinae) [14]. The latter subfamily, however, is unable to produce bufadienolides autogenously but sequesters them by preying on species of *Photinus* (Lampyrinae). Female *Photuris* imitate the flash signal characteristics of *Photinus* females and attract male *Photinus* that are subsequently eaten [14]. It has been suggested that the ability to biosynthesize bufadienolides is restricted to the Lampyrinae but, so far, *Aquatica lateralis* from the subfamily Luciolinae is the only species that has been investigated outside of the Lampyrinae [24].

In fireflies, bufadienolides mediate defense [9], and fireflies were repeatedly shown to be protected against various predators including vertebrates and invertebrates [14,25,26]. Like cardenolides [27], firefly bufadienolides are emetic to vertebrates [9,28], but their physiological effects on invertebrate predators are unknown. Many species show reflex bleeding, i.e., exudation of lucibufagin-containing hemolymph from the coxal joints, pronotum, and elytra upon tactual stimulation [16,29]. Remarkably, lucibufagins were found in all developmental stages of fireflies, i.e., eggs, larvae, and pupae [16]. In addition to its role as a courtship signal, light emission in fireflies was also shown to function as an aposematic signal for advertising toxicity to predators [30,31]. Unpalatability is furthermore advertised to bats by the production of ultrasonic signals, as observed in four species of the subfamily Luciolinae [32] for which the nature of their chemical defenses is unclear.

In our proof-of-concept study, we screened dry museum specimens of 31 palearctic firefly species and 12 specimens of the related aposematic net-winged beetles (Lycidae) for the presence of bufadienolides using a non-invasive methanolic extraction process and HPLC-DAD and HPLC-MS for analysis. Cardenolides are known to be comparatively stable—a property most likely also applying to the structurally closely related bufadienolides. Moreover, bufadienolides have melting points >200 ℃ [33], indicating very low volatility under standard conditions. Given their comparatively high chemical stability and non-volatility, firefly bufadienolides represent suitable candidate molecules for our approach. We tested the hypothesis that the production of lucibufagins is restricted to genera of the subfamily Lampyrinae, as was suggested by Fallon et al. [24]. Our approach may help to reconstruct the evolution of bufadienolide synthesis within the Lampyridae. Further, we tested whether our non-invasive screening approach in museum specimens is useful for drug discovery in natural history collections.

## 2. Materials and Methods

### 2.1. Non-Destructive Extraction of Dry Museum Specimens

We tested 60 lampyrid specimens of the genera *Lamprohiza* (4 species), *Lampyris* (13 species), *Lampyroidea* (3 species), *Luciola* (3 species), *Nyctophila* (6 species), *Pelania* (1 species), and *Phosphaenus* (1 species) for the occurrence of lucibufagins, only one of which (*L. noctiluca*) has been studied before [18]. In addition, we analyzed 6 species of the closely related Lycidae (11 specimens). Specimens were briefly soaked in tap water for removal from glue boards. Subsequently, they were air-dried, and immersed in 1 mL of methanol (Rotipuran ≥ 99.9%, Carl Roth, Karlsruhe, Germany). After 24 h at room temperature, the supernatant was transferred to a fresh tube and the extraction was repeated twice with methanol. Pooled supernatants (three in total) were evaporated under a flow of cold N_2_. Dry residues were dissolved in 100 µL (300 µL for a couple of samples) of methanol by agitation in a FastPrep-24^TM^ 5G instrument (Biomedicals GmbH, Eschwege, Germany) for two 45 s cycles at 6.5 m/s. Following centrifugation (16,100× *g*, 3 min), extracts were applied to unconditioned solid-phase extraction columns (Chromabond, 200 mg SiOH normal phase, Macherey-Nagel, Düren, Germany) that were subsequently eluted with 10 mL of n-hexane (Rotisolv ≥ 95%, Carl Roth, Karlsruhe, Germany) to remove lipophilic compounds. Next, the columns were eluted with 6 mL of MeOH, and the resulting extracts were evaporated under N_2_. Dry residues were dissolved in 100 µL of methanol by 45 s of vortexing. After filtration (Rotilabo^®^ nylon syringe filters, Ø 13 mm, 0.45 µm pore size, Carl Roth, Karlsruhe, Germany), extracts were subjected to DAD-HPLC (diode array detector-HPLC) and DAD-LC/ESI-Qq-TOF-mass spectrometry (electrospray ionization quadrupole-quadrupole time-of-flight mass spectrometry). In total, we obtained extracts from 72 coleopteran specimens. Besides fireflies, we included six species of net-winged beetles (Lycidae) which are closely related to the Lampyridae and possess a red aposematic coloration, indicating the occurrence of chemical defenses; see Appendix A for a complete list of samples. After extraction and evaporation of solvents, beetle specimens were remounted and retransferred to collections (see Appendix A). We used dried *P. pyralis* obtained from Sigma-Aldrich (Taufkirchen, Germany) as a positive control for the development of our methods. 

### 2.2. DAD-HPLC and DAD-LC/ESI-Qq-TOF-mass Spectrometry of Beetle Extracts

For lucibufagin-screening and relative quantification, 15 µL of extract were injected into an Agilent 1100 series instrument (Agilent, Waldbronn, Germany) and compounds were separated on an EC 150/4.6 Nucleodur^®^ C_18_ Gravity column (3 μm, 150 mm × 4.6 mm, Macherey-Nagel, Düren, Germany) at a constant flow of 0.7 mL/min using the following acetonitrile-H_2_O gradient: 0–2 min 16% acetonitrile, 2–25 min 70% acetonitrile, 25–30 min 95% acetonitrile, 30–35 min 95% acetonitrile, reconditioning for 2 min at 16% acetonitrile and 10 min equilibration at 16% acetonitrile. UV absorbance spectra were recorded from 200 to 400 nm with a diode array detector, and data evaluation was carried out using the ChemStation software (Rev. B. 04.03.16, Agilent Technologies, Waldbronn, Germany). Lucibufagin peaks were identified based on their symmetrical absorption maximum between 295 and 302 nm [11,13]. Areas of lucibufagin peaks were compared to an external calibration curve of the bufadienolide proscillaridin A (PhytoLab, Vestenbergsgreuth, Germany; 5, 10, 25, 50, 100, 250, and 500 µg/mL) at 300 nm for quantification.

A subset of extracts containing lucibufagins based on DAD-HPLC assessment was subjected to DAD-LC/ESI-Qq-TOF-mass spectrometry for tentative structural elucidation of lucibufagins. For this purpose, samples were diluted by adding 85 µL of methanol each. Ten microliters of extract were injected into an UltiMate 3000-HPLC (Dionex, Idstein, Germany). Compounds were separated on a Kinetex C_18_-column (2.6 µm, 150 × 2.1 mm, Phenomenex, Aschaffenburg, Germany) using a gradient of acetonitrile and water (both containing 0.1% formic acid) as follows: 0–2 min 5% acetonitrile, 2–20 min 95% acetonitrile, 20–24 min 95% acetonitrile, reconditioning for 1 min at 5% acetonitrile and 5 min equilibration at 5% acetonitrile. We ran a 5 µL methanol blank to clean the system of potential impurities after each sample. High-resolution mass spectra were recorded on a micrOTOF Q-II mass spectrometer using the HyStar v. 3.2 SR4 software (Build 49.9; Bruker Daltonic, Bremen, Germany). Data were recorded using the otof Control Version 3.4 software (Build 14; Bruker Daltonic, Bremen, Germany). Samples were screened for most masses of lucibufagins that have been described to occur in fireflies previously [10,11,12,13,15,16]. Only pseudomolecular ions (exclusively [M+H]^+^) with a relative intensity of > 6 × 10^4^ in the MS signal (TIC) were subjected to subsequent MS/MS analysis.

## 3. Results and Discussion

We detected lucibufagins in 21 European lampyrid species of the genera *Lampyris*, *Nyctophila*, *Pelania*, and *Phosphaenus* as well as in the North-African species *Lampyris algerica* (subspecies *levigata*), all belonging to the subfamily Lampyrinae (Figure 1). *Lampyris noctiluca*, which had been reported as a source of lucibufagins previously [18], was confirmed as a bufadienolide-containing species (see Figure 1 and Table 1). In contrast, we found no lucibufagins in 10 species of the lampyrid genera *Lampyroidea*, *Luciola*, and *Lamprohiza* belonging to the subfamilies Luciolinae and Lamprohizinae, respectively. Our species collection comprised 13 out of 16 known species of *Lampyris* (www.biolib.cz, accessed on 25 May 2021), and 6 out of 13 described species of *Nyctophila* (www.biolib.cz; *Pelania* and *Phosphaenus* are monotypic genera). Since we found lucibufagins in all tested species of *Lampyris* and *Nyctophila*, it seems likely that production of bufadienolides is shared across all species of these two genera. Similarly, all investigated species of *Lamprohiza* (four species), *Lampyroidea* (three species), and *Luciola* (three species) were devoid of the compounds suggesting that the ability to produce lucibufagins is not a basal trait of the Lampyridae and arose at the basis of the subfamily Lampyrinae (Figure 1). In the Photurinae (*Photuris* spp.), lucibufagins are acquired from predation on fireflies of the genus *Photinus* and are not synthesized autogenously [14]. Therefore, it may well be the case that the ability to produce lucibufagins is restricted to the subfamily Lampyrinae, as was suggested by Fallon et al. [24].

In total, we screened 45 specimens of the genera *Lampyris*, *Nyctophila*, *Pelania*, and *Phosphaenus* and found lucibufagins in all individuals (although some specimens had very low amounts, see Table 1), suggesting a high reliability of our assessment based on museum specimens. From the genera *Lamprohiza*, *Lampyroidea*, and *Luciola*, 15 specimens were screened, all of which were devoid of bufadienolides, making it unlikely that the compounds were overlooked (see Appendix A for details on the species). We did not detect bufadienolides in 12 specimens belonging to the family Lycidae (6 species), which are closely related to the Lampyridae [34], supporting the hypothesis that the occurrence of lucibufagins is restricted to the fireflies.

Across our firefly samples, we found five pseudomolecular ions ([M+H]^+^) representing five individual lucibufagins that have not been described before. Specifically, the following [M+H]^+^ ions were found: *m/z* 529.2073, *m/z* 575.2496, and *m/z* 593.2568 from *Lampyris noctiluca* (sample IDs 22, 24, 24), as well as *m/z* 549.2340 from *L. hellenica* (sample ID48) and *m/z* 549.2666 from *L. brutia* (sample ID68). Notably, the latter two are of the same nominal mass, but our high-resolution data clearly indicate that two lucibufagins with distinct molecular formulae are present. This emphasizes the analytical strength of high-resolution data for electrospray mass spectrometry screening (as outlined below in the Results and Discussion section). Due to the limited amount of lucibufagin-containing extracts, only tentative structures of these unknown compounds can be proposed as follows:

The new compound with the pseudomolecular ion *m/z* 529.2073 ([M+H]^+^) from *L. noctiluca* is a homolog of lucibufagin C. The latter has also been found in the same sample (*m/z* 533.2366 [M+H]^+^, calcd. for C_28_H_37_O_10_: 533.2387). The new compound obviously contains two additional double bonds. This hypothesis corresponds well with its molecular weight *m/z* 529.2074, calcd. for C_28_H_33_O_10._ However, the position of these double bonds cannot be assigned. Currently, 4,12-dioxo-3α-oxylosyl-9,11-en-11-hydroxybufalin from *Lucidota atra* is the only lucibufagin carrying an additional 9,11-double bond in the steroid core [15].

A new compound from *L. hellenica* displaying its pseudomolecular ion *m/z* 549.2340 ([M+H]^+^) is not an unsaturated homolog of lucibufagin B (calcd. for C_29_H_39_O_10_), that only possesses one double bond. Instead, our HR-MS data suggest the molecular formula C_28_H_37_O_11_, which is in excellent agreement with the observed pseudomolecular ion *m/z* 549.2336 ([M+H]^+^). Formally, this new metabolite found in our study contains an additional carbonyl group, which could be attached to the cyclopentane ring, and only one double bond in the δ-lactone ring. 

The pseudomolecular ion *m/z* 549.2666 ([M+H]^+^) belongs to a new lucibufagin found in *L. brutia*. This compound formally represents an unsaturated homolog (*m/z* 549.2700, calcd. for C_29_H_41_O_10_) of lucibufagin B, displaying only one double bond in the δ-lactone ring. However, this hypothesis could not be confirmed due to the low intensity of the [M+H]^+^ ion that should be used as the precursor for further MS/MS experiments. Remarkably, our sample of *L. brutia* was collected in 1905; and we extracted lucibufagins from several specimens that also were older than 100 years (the oldest specimen was collected in 1899). Consequently, even very old museum specimens can reveal important information to address biological questions and lead to the discovery of novel natural products. Although bufadienolides likely represent comparatively stable compounds, it is known that some cardiotonic steroids are sensitive to autoxidation [35,36]. Therefore, we cannot completely exclude that the previously undescribed compounds might represent degradation products formed during preparation, conservation, or long-term storage of the museum specimens. Consequently, comparisons to freshly collected specimens will be necessary to validate the occurrence of the novel bufadienolides in living fireflies. The amounts of bufadienolides in some of the oldest specimens were among the lowest amounts found overall, which may indicate degradation of lucibufagins over time. In contrast, we found comparatively high amounts of bufadienolides in *L. brutia* from 1905 and *N. molesta* from 1910. Consequently, it will require further research involving higher numbers of specimens to understand the influence of storage time on bufadienolide content.

In a sample from *L. noctiluca*, we found two major lucibufagins, *m/z* 575.2496 and *m/z* 593.2568 ([M+H]^+^). The former is another homolog of lucibufagin C, which is characterized by a third AcO substituent. This hypothesis agrees very well with the calculated *m/z* 575.2492, corresponding to the molecular formula C_30_H_39_O_11_ ([M+H]^+^). Further MS/MS experiments clearly indicated the triple loss of AcOH. However, the position of the additional AcO group, which replaces one of the free OH groups, could not be assigned. Compared to *m/z* 575.2496, the latter compound carries an additional OH group. This hypothesis is corroborated by HR-MS data, confirming the calculated pseudomolecular ion *m/z* 593.2598, which corresponds to the molecular formula C_30_H_41_O_12_ ([M+H]^+^).

Besides the five novel lucibufagins, we found the following pseudomolecular ions [*m*/*z*]: 433, 435, 447, 449, 477, 491, 505, 507, 529, 533, 547, 549, 561, 563, 565, 575, and 593 (all [M + H]^+^; see Appendix A for the occurrence in individual species) representing firefly lucibufagins that were described previously [10,11,12,13,15,16]. Due to the comparatively low intensities of these ions (<6 × 10^4^ in the total ion current of the MS), no fragmentation experiments (i.e., MS/MS) could be performed. This list, however, most likely is not complete, as indicated by the high number of different bufadienolide peaks detected by our assessment based on DAD (Figure 1). Gronquist et al. [15] found 13 different lucibufagins in *L. atra* and it may be common that fireflies contain structurally diverse arrays of lucibufagins. Here, we found up to 38 different peaks with DAD-spectra exhibiting the characteristics of bufadienolides (Table 1). While the assessment by DAD is prone to some extent of misidentification and some of the bufadienolides found may represent degradation products due to the use of old museum specimens, it seems likely that many uncharacterized bufadienolides in fireflies exist. Smedley et al. reported >500 µg bufadienolides per individual in *E. corrusca* [16], which is in good agreement with the highest amounts that we determined here (ca. 670 µg/beetle in *Lampyris hellenica*).

Our study confirms that a phylogeny-guided screening for natural products is informative in attributing evolutionary novelties in terms of natural products to particular taxa [37]. While the use of dried museum specimens seems very promising for non-volatile, chemically stable molecules, its suitability for volatile and chemically labile substances will need to be addressed in future studies. Besides bufadienolides, we successfully extracted and analyzed cardenolides as well as different classes of alkaloids from dried specimens of Heteroptera ([7], Petschenka unpublished data), suggesting that our approach will be suitable for a wide array of chemical compounds.

The most parsimonious interpretation of the observed pattern of lucibufagin occurrence across the lampyrid species sampled places the ability to produce lucibufagins at the basis of the subfamily Lampyrinae. Alternatively, lucibufagin synthesis could represent a synapomorphy of the subfamilies Psilocladinae, Amydetinae, Photurinae, and Lampyrinae, which would require a loss of the ability to produce bufadienolides in the Photurinae, and therefore is less likely. To better understand the evolution of bufadienolide production in the Lampyridae, it would be very informative to analyze specimens of the subfamilies Psilocladinae and Amydetinae (see Figure 1). The presence of bufadienolides (if not also acquired by predation as seen in the Photurinae) in the Psilocladinae or both, the Psilocladinae and Amydetinae, would support a loss of bufadienolide production in the common ancestor of Photurinae and Amydetinae or the Photurinae, respectively.

Finally, we demonstrated that non-invasive methanol extraction of museum specimens and the analysis of the extracts using HPLC-DAD and HPLC-MS enables the discovery of new natural products, making this approach promising for large-scale screenings for natural products in natural history collections. Our method is non-invasive, i.e., the extracted beetles were not damaged by the process and mostly looked even cleaner (see Appendix A) when returned to the museum. In conclusion, screening for natural products based on natural history collections is not necessarily in conflict with preserving valuable specimens.

## Figures and Tables

**Figure 1 insects-12-00689-f001:**
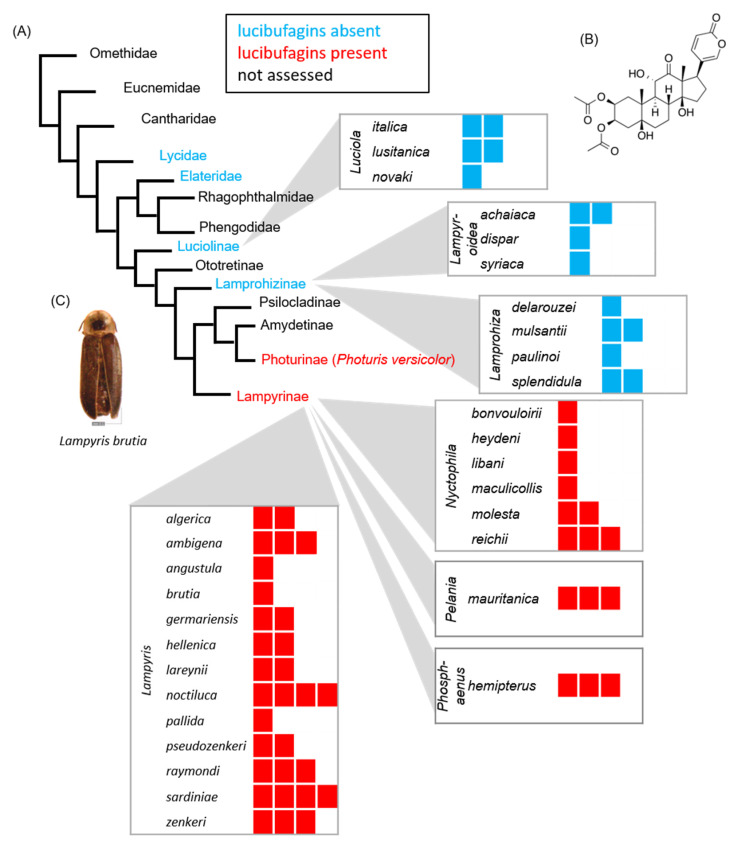
Distribution of lucibufagins (i.e., bufadienolides) in fireflies based on the chromatographic analysis of museum specimens. (**A**) 31 palearctic species of fireflies (Lampyridae) were assessed for the occurrence of defensive lucibufagins. Red rectangles represent beetle specimens that tested positively for lucibufagins, blue rectangles represent beetle individuals that were devoid of these compounds. Further occurrences of bufadienolides are known from *Photuris versicolor* (Photurinae) [14], *Diaphanes lampyroides* [17], *Ellychnia corrusca* [16], *Lucidota atra* [15], *Photinus ignitus, P. marginellus* [9,12], and *P. pyralis* (Lampyrinae) [11,12]. Absence of lucibufagins was further demonstrated for *Aquatica lateralis* (subfamily Luciolinae) and the click beetle *Ingnelater luminosus* (Elateridae) [24]. Coloration of family or subfamily names either indicates presence (red) or absence (blue) of bufadienolides based on literature research and our own evaluation. Phylogenetic relationships are based on Martin et al. [34]. (**B**) Lucibufagin C, a firefly bufadienolide found in *Photinus pyralis* [11]. (**C**) Extracted specimen of *Lampyris brutia*, collected in 1905.

**Table 1 insects-12-00689-t001:** Total amounts of lucibufagins and diversity of structurally different lucibufagins based on HPLC-DAD.** Individual lucibufagin contents are separated by slashes if more than one specimen was analyzed (second column). Diversities of individual lucibufagins (third column) are listed in the same manner. Note that only positively screened species are listed in the table.

Species	Total Lucibufagins per Beetle [µg]	Number of Lucibufagin Peaks
*Lampyris algerica levigata* (Geisthardt 1983)	154/165	24/25
*Lampyris ambigena* (Jacquelin du Val 1860)	7/63/185	9/20/22
*Lampyris angustula* (Fairmaire 1895)	41	21
*Lampyris brutia* (Costa 1882)	123	24
*Lampyris germariensis* (Jacquelin du Val 1860)	3/60	6/18
*Lampyris hellenica* (Geisthardt 1983)	114/668	24/41
*Lampyris lareynii* (Jacquelin du Val 1859)	29/78	19/21
*Lampyris noctiluca* (Linnaeus 1767)	88/118/142/615	24/14/22/30
*Lampyris pallida* (Geisthardt 1987)	34	23
*Lampyris pseudozenkeri* (Geisthardt 1999)	11/19	21/21
*Lampyris raymondi* (Mulsant and Rey 1859)	5/133/313	6/21/24
*Lampyris sardiniae* (Geisthardt 1987)	2/77/125/138	4/26/38/32
*Lampyris zenkeri* (Germar 1817)	53/114/210	14/21/16
*Nyctophila heydeni* (Olivier 1884)	55	11
*Nyctophila maculicollis* (Fairmaire 1866)	102	19
*Nyctophila molesta* (Jacquelin du Val 1859)	291/323	30/22
*Nyctophila reichii* (Jacquelin du Val 1859)	56/217/309	11/28/21
*Pelania mauritanica* (Linnaeus 1767)	64/159/194	16/22/25
*Phosphaenus hemipterus* (Goeze 1777)	9/18/38	18/24/29

## Data Availability

All data underlying the manuscript are presented in this manuscript and the supplementary material.

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
