# Peer review of "Insect Collections as an Untapped Source of Bioactive Compounds—Fireflies (Coleoptera: Lampyridae) and Cardiotonic Steroids as a Proof of Concept"

_insects, 2021, doi:10.3390/insects12080689_

Round 1
Reviewer 1 Report
Review
Insect Collections as an Untapped Source of Bioactive Compounds – Fireflies (Coleoptera: Lampyridae) and Cardiotonic Steroids as a Proof of Concept is a very interesting, well-executed, and well-written report on the potential of non-invasive, non-destructive use of museum material to advance investigations of ecological interactions from past to present. In conjunction with phylogenetic analyses, this methodology shows great promise.
Minor comments
Intro, l2: Although vertebrates, such as bats and birds, are among the chief predators of fireflies, it should be mentioned that these cardiotonic toxins are also emetics and are also potent against invertebrate enemies because they affect many other homeostatic physiological processes that depend on the K/Na pump.
Intro: It might be pointed out that cardenolides also have one double bond, while bufodienolides have two in the lactone ring. As far as nomenclature is concerned, might it not be simpler to class them as two groups within the cardiotonic glycosides or as cardiotonic steroids?
Spell out HPLC-DAD HPLC-MS at first mention in Methods.
The authors might site Zangerl & Berenbaum for their pioneering use of museum material for investigations of chemically mediated ecological interactions: Zangerl & Berenbaum (2005). Increase in toxicity of an invasive weed after reassociation with its coevolved herbivore. PNAS October 25, 102 (43) 15529-15532. An emphasis on the non-destructive impact of the present method might be added here for support of the concept.
Mention is made of the study (ref 28) describing the use of ultrasonic wing clicking in some SE Asian fireflies as an aposematic warning. It is important to also note that the chemical basis for adult firefly unprofitability has not been established for Asian adult fireflies. These species are in the Luciolinae.
p7: Can anything more specific be said about quantitative recovery? Can you compared the yields ug/mg between fresh and curated? I would suspect these compounds as a group to be quit stable.
In the supplemental table, the authors might add subfamily as the third column.
Author Response
Thanks a lot for your thorough assessment of our article! Please see below for a point-by-point response to all comments.
Intro, l2: Although vertebrates, such as bats and birds, are among the chief predators of fireflies, it should be mentioned that these cardiotonic toxins are also emetics and are also potent against invertebrate enemies because they affect many other homeostatic physiological processes that depend on the K/Na pump.
Thank you for your suggestion! We added “including vertebrates and invertebrates” to our sentence in line 99. One of the references cited actually refers to jumping spiders. Furthermore, we added that bufadienolides possess emetic properties (line 100).
Intro: It might be pointed out that cardenolides also have one double bond, while bufodienolides have two in the lactone ring. As far as nomenclature is concerned, might it not be simpler to class them as two groups within the cardiotonic glycosides or as cardiotonic steroids?
We added this information in line 75-76. Yes, cardenolides and bufadienolides both belong to the cardiotonic steroids or cardiac glycosides, see definition in line 72-73.
Spell out HPLC-DAD HPLC-MS at first mention in Methods.
We spelled out "DAD" and "DAD-LC/ESI-Qq-TOF" in line 145-147. To avoid a stringing of too many nouns, we did not spell out HPLC and LC-MS since we believe that these abbreviations are so commonly used that scientific readers know what they mean.
The authors might site Zangerl & Berenbaum for their pioneering use of museum material for investigations of chemically mediated ecological interactions: Zangerl & Berenbaum (2005). Increase in toxicity of an invasive weed after reassociation with its coevolved herbivore. PNAS October 25, 102 (43) 15529-15532. An emphasis on the non-destructive impact of the present method might be added here for support of the concept.
Thank you very much for this suggestion! We cited Zangerl & Berenbaum, and also two other studies analyzing chemicals derived from museum specimens (line 56). We also slightly changed the wording in our manuscript (line 29, 56) not to cause the impression that we were the first to analyze museum specimens (although all other studies had a different motivation).
Mention is made of the study (ref 28) describing the use of ultrasonic wing clicking in some SE Asian fireflies as an aposematic warning. It is important to also note that the chemical basis for adult firefly unprofitability has not been established for Asian adult fireflies. These species are in the Luciolinae.
We added this information in line 108-110.
p7: Can anything more specific be said about quantitative recovery? Can you compared the yields ug/mg between fresh and curated? I would suspect these compounds as a group to be quit stable
Unfortunately, we did not carry out quantitative comparisons between fresh and dry specimens due to the lack of material. We agree and added some information regarding the stability of bufadienolides (line 114-119).
In the supplemental table, the authors might add subfamily as the third column.
Good suggestion! We added a column stating the subfamily.
Reviewer 2 Report
Review insects-1307065
Insect Collections as an Untapped Source of Bioactive ComPounds – Fireflies (Coleoptera: Lampyridae) and Cardio-Tonic Steroids as a Proof of Concept
This manuscript has two aims: (1) demonstrating that insect collections provide a rich source for discovering bioactive compounds, and (2) using fireflies, specifically lucibufagins as an example how such data can be used in evolutionary studies. Both aims provide a highly valuable contribution to the literature and should be of great interest to the readers of this journal. However, to fully achieve the stated aims a few improvements should be made to the manuscript before publication.
For example, the journal Insects targets entomologists all over the world with diverse interests and backgrounds. While some readers may be specialists in chemistry and chemical analyses, this is likely not the case for the large majority of the readers. Therefore, additional context and explanations on terminology and methods are required. In addition, a section on possible sources of error and on the suitability of this approach for different chemical compounds should be added.
Similarly, the phylogenetic hypothesis testing requires some elaboration. A hypothesis should be a clearly defined testable explanation and alternative hypotheses should be discussed based on the findings, parsimony and sampling used (for suggestions please see below).
Specific comments and suggestions:
Page 1:
- “ a non-destructive approach, we screened 72 dry coleopteran specimens for bufadienolides using HPLC-DAD and HPLC-MS. We found bufadienolides including five novel compounds in 21 species of the subfamily Lampyrinae. “
Comments: Please add some more background. For example, how stable is this group of chemicals? Is any denaturation possible? Either through preparation, conservation, or long-term storage of specimens? Does this impact only the quantity of detected compounds? Or can this generate “new variants”? An explanation is needed here to put this approach in context, e.g. why lucibufagins are good bioactive compounds to do this study with.
This should be followed up in the results/discussion with an assessment of how confident we can be that the detected novel compounds/variants (e.g. novel compounds and novel pseudomolecular ions) are not products of denaturation.
In addition (given the title/aim of this manuscript), a discussion dedicated to differences between different bioactive chemical compounds and their potential to be discovered in museum specimens (i.e. survive preparation, conservation, or long-term storage of specimens) should be added to the discussion section of this manuscript.
Page 3:
- “We tested the hypothesis that the occurrence of lucibufagins shows a phylogenetic pattern, which may help to reconstruct the evolution of bufadienolide synthesis within the Lampyridae”
Comment: With respect to the stated sentence, I am not sure what the hypothesis (testable explanation) on the phylogenetic pattern is. The pattern is never explicitly stated. Also please note that occurrence versus synthesis is mixed in this sentence, and we already know from Photuris (page 2) that occurrence data would introduce error to the phylogenetic reconstruction of the origin of synthesis.
Suggestions:
Given the extensive sampling of species within genera in this study it would make sense to explicitly add this as a contribution, e.g. testing whether the occurrence of lucibufagins is consistent within genera. This is especially important since this study also adds more evidence (to existing literature data) that additional clades/likely all clades? within Lampyrinae may have lucibufagins.
To develop the phylogenetic connection, one possibility would be to build on the genus data by generating a hypothesis on the phylogenetic origin of lucibufagin synthesis in fireflies (explaining the observed patterns of presence and absence of lucibufagins across genera in this study and the literature). This hypothesis could be tested with additional data (additional genera or sampling within genera) in the future. In any case, a phylogenetic approach to the lucibufagin data requires a more thorough discussion of the phylogeny and implications for lucibufagin production in fireflies.
- “Further, we tested whether our noninvasive screening approach in museum specimens is useful for drug discovery in natural history collections”
Comments: This is an exciting possibility, however this statement should be followed up with a more detailed discussion (in discussion section) about which kind of chemical compounds/drugs (major classes) would lend themselves to this and which others are less suitable (e.g. denaturation due to chemical or light-induced processes).
Methods section:
Please list the genera analyzed and sample sizes here (not just in results) and state which taxa are new to literature to emphasize the new contributions here. The sampling is very impressive, but this gets lost in the present version.
Page 4:
- “Lucibufagin peaks were identified based on their symmetrical absorption maximum between 295-302 nm (Meinwald 1979, González et al. 1999b).”
Comment: Are lucibufagins the only compounds with a symmetrical absorption spectrum in that range? Could whole-body extracts (rather than pure materials as in Meinwald) contain other compounds in this range? This is unclear and should be briefly explained.
Also please note: the citations style used here (and in methods section below) is author, year rather than a number as in the remainder of manuscript.
- “Only pseudomolecular ions with a relative intensity of > 6 Í 104 in the MS signal (TIC) were subjected to subsequent MS/MS analysis.”
Comment: Please define what pseudomolecular ions are.
- “Similarly, all investigated species of Lamprohiza (four species), Lampyroidea (three species), and Luciola (three species) were devoid of the compounds suggesting that the ability to produce lucibufagins …. arose before the diversification of the clade comprising the subfamilies Lampyrinae and Photurinae (Fig. 1).”
Comment: This is not the most parsimonious conclusion with respect to lucibufagin production as shown here (and using the Martin et al. 2019 phylogeny):
- Since Photurinae do not synthesize their own lucibufagins (but acquire them through Photinus predation) as the authors themselves state in the next sentence, the production of lucibufagins can only be placed at the base of the Lampyrinae branch (requires 1 evolutionary event).
- There is the alternative - and less likely/less parsimonious - scenario (requires 2 events) that the ability to produce lucibufagins originated in the common ancestor of Psillocladinae, Amydetinae, Photurinae and Lampyrinae and was subsequently lost in Photurinae, e.g. due their ability to acquire them through predation. However, testing this second scenario requires data for Psilocladinae and Amydetinae, and to support the second scenario requires the presence of lucibufagins in at least Psilocladinae (i.e. lucibufagin loss in the ancestor of Photurinae and Amydetinae) or both Psilocladinae and Amydetinae (i.e. lucibufagin loss in Photurinae).
Since Psillocladinae and Amydetinae were not included in the study, the authors should at least discuss these two scenarios as alternative hypotheses on the origin of lucibufagin production (and the reasoning behind them) here. This would also give the conclusion sentence (page 7) “To understand the evolution of bufadienolide production in the Lampyridae it will be very informative to analyze specimens of the subfamilies Ototretinae, Psilocladinae, and Amydetinae” context for the reader and more specifics to actually follow the reasoning behind this statement; please add what Ototretinae data would add/what question it would answer.
- Therefore, it may well be the case that the ability to produce lucibufagins is restricted to the subfamily Lampyrinae as was suggested by Fallon et al. [22].
Comment: This may go beyond the scope of the present manuscript, but if this is the case: what does this mean for origin of lucibufagin synthesis? Is this still an important question? (if yes, which other taxa within Lampyrinae should be tested to identify the origin?)
Page 6:
- “We did not detect bufadienolides in 12 specimens belonging to the family Lycidae (6 species), which are closely related to the Lampyridae [29] supporting the hypothesis that the occurrence of lucibufagins is restricted to the fireflies.”
Comment: What about Rhagophthalmidae and Phengodidae, two beetle families that are more closely related to fireflies than lycids?
- Table 1: Total amounts and diversity of structurally different lucibufagins
Comment: The data in this table are never presented or discussed in detail (in results or in discussion): what does the individual variation in total lucibufagins imply for using collections for detecting bioactive compounds (e.g. minimum specimen required for analysis and/or evidence of denaturation)? How confident can we be that these are functional compounds given this variation? Is this also typical for “fresh” specimen? Same for number of lucibufagin peaks? How can this inter-individual variation be interpreted (on its own and compared to total) with respect to extracting bioactive compounds (of this type) from specimens in NH collections?
- “Across our firefly samples, we found four pseudomolecular ions “
Comment: Please be more specific (and define terminology in methods) and explain what this is in the context of this study. Are these indicators for lucibufagins/candidates? Or false positives? It would be helpful to set the stage on how to interpret these before readers engage with the rest of the paragraph.
Page 7:
- Remarkably, our sample of L. brutia was collected in 1905; and we extracted lucibufagins from several specimens that also were older than 100 years (the oldest specimen was collected in 1899). Consequently, even very old museum specimens can reveal important information to address biological questions and lead to the discovery of novel natural products.
Comment: This is amazing! In terms of what we can learn from them: Is there any suggestion in total number of lucibufagins and peaks of the older specimens with respect to conservation/denaturation of lucibufagins?
- “Besides the four putatively novel lucibufagins, we found the following putative pseudomolecular ions ..”
Comment: Please explain (in methods and here). What is the difference between:
- a putatively novel lucibufagins and pseudomolecular ions?
- a putatively novel lucibufagins and putative pseudomolecular ions?
Please explain in the methods in more detail (conceptually) what these categories are and which criteria were used to decide between these categories.
- To understand the evolution of bufadienolide production in the Lampyridae it will be very informative to analyze specimens of the subfamilies Ototretinae, Psilocladinae, and Amydetinae (see Fig. 1).
Comment: Please expand reasoning (see above)
- Finally, we demonstrated that non-invasive methanol extraction of museum specimens and the analysis of the extracts using HPLC-DAD and HPLC-MS enables the discovery of new natural products, making this approach promising for large-scale screenings for natural products in natural history collections.
Comment: Please add a discussion on sources of error (see above) and how to verify products versus artifacts. Also please add how generalizable these findings on lucibufagins/cardiac glycosides are for other chemical compounds (see above).
Author Response
Thanks a lot for your thorough assessment of our article! Please see below for a point-by-point response to all comments.
“ a non-destructive approach, we screened 72 dry coleopteran specimens for bufadienolides using HPLC-DAD and HPLC-MS. We found bufadienolides including five novel compounds in 21 species of the subfamily Lampyrinae. “
Comments: Please add some more background. For example, how stable is this group of chemicals? Is any denaturation possible? Either through preparation, conservation, or long-term storage of specimens? Does this impact only the quantity of detected compounds? Or can this generate “new variants”? An explanation is needed here to put this approach in context, e.g. why lucibufagins are good bioactive compounds to do this study with
Thank you for your suggestion! We added some information regarding the stability of bufadienolides to the manuscript (lines 114-119, 278-289) and discussed the potential of degradation (lines 278-288, lines 309-311). We cannot exclude that the newly discovered bufadienolides represent degradation products. From a pharmacognostic perspective, however, they still represent novel compounds.
This should be followed up in the results/discussion with an assessment of how confident we can be that the detected novel compounds/variants (e.g. novel compounds and novel pseudomolecular ions) are not products of denaturation.
Done, please see above.
In addition (given the title/aim of this manuscript), a discussion dedicated to differences between different bioactive chemical compounds and their potential to be discovered in museum specimens (i.e. survive preparation, conservation, or long-term storage of specimens) should be added to the discussion section of this manuscript.
We agree but we could only speculate about the properties of different chemical classes. This will need to be tested empirically. We added some examples of other classes of compounds that we succesfully analyzed from museum specimens to corroborate our approach (lines 319-322).
“We tested the hypothesis that the occurrence of lucibufagins shows a phylogenetic pattern, which may help to reconstruct the evolution of bufadienolide synthesis within the Lampyridae”
Comment: With respect to the stated sentence, I am not sure what the hypothesis (testable explanation) on the phylogenetic pattern is. The pattern is never explicitly stated. Also please note that occurrence versus synthesis is mixed in this sentence, and we already know from Photuris (page 2) that occurrence data would introduce error to the phylogenetic reconstruction of the origin of synthesis.
Thank you for pointing this out! We specified our hypothesis (line 119-120). We also used "prodcution" instead of "occurence" to clarify.
Suggestions:
Given the extensive sampling of species within genera in this study it would make sense to explicitly add this as a contribution, e.g. testing whether the occurrence of lucibufagins is consistent within genera. This is especially important since this study also adds more evidence (to existing literature data) that additional clades/likely all clades? within Lampyrinae may have lucibufagins.
Thank your for this suggestion. We are not sure if we correctly understand what is meant here... Are you suggesting a statistical comparison in the sense of a correlation analysis between genus and the occurence of bufadienolides? Since we see a clear "presence/absence" pattern across genera we are unsure about the value of such an analysis... I'm worried we did not really get your point?
To develop the phylogenetic connection, one possibility would be to build on the genus data by generating a hypothesis on the phylogenetic origin of lucibufagin synthesis in fireflies (explaining the observed patterns of presence and absence of lucibufagins across genera in this study and the literature). This hypothesis could be tested with additional data (additional genera or sampling within genera) in the future. In any case, a phylogenetic approach to the lucibufagin data requires a more thorough discussion of the phylogeny and implications for lucibufagin production in fireflies.
We added an improved discussion and formulated hypotheses about the origin of lucibufagins (lines 323-334). We agree that a broader sampling including additional subfamilies would be very interesting!
“Further, we tested whether our noninvasive screening approach in museum specimens is useful for drug discovery in natural history collections”
Comments: This is an exciting possibility, however this statement should be followed up with a more detailed discussion (in discussion section) about which kind of chemical compounds/drugs (major classes) would lend themselves to this and which others are less suitable (e.g. denaturation due to chemical or light-induced processes).
We added some discussion and examples of other classes of compounds that we succesfully analyzed from museum specimens to corroborate our approach (lines 317-322), please see above.
Methods section:
Please list the genera analyzed and sample sizes here (not just in results) and state which taxa are new to literature to emphasize the new contributions here. The sampling is very impressive, but this gets lost in the present version.
We included the information in the methods section (line 127-131).
“Lucibufagin peaks were identified based on their symmetrical absorption maximum between 295-302 nm (Meinwald 1979, González et al. 1999b).”
Comment: Are lucibufagins the only compounds with a symmetrical absorption spectrum in that range? Could whole-body extracts (rather than pure materials as in Meinwald) contain other compounds in this range? This is unclear and should be briefly explained.
This is a general issue of DAD-analyses... The issue was alreadey mentioned in line 309-310. Since peaks with typical bufadienolide spectra are absent outside of the Lampyrinae and in the Lycids it is very unlikely that they don't represent bufadienolides (i.e. there were no dominant ubiquitous compounds in our samples showing a similar spectrum). Moreover, our results are validated with LC-MS measurements which unambiguously proof our compounds being bufadienolides. We did not add additional explanation except of what is mentioned in line 309-310 since some potential of misinterpretation (which seems very unlikely in our case) is intrinisc to DAD-measurements.
Also please note: the citations style used here (and in methods section below) is author, year rather than a number as in the remainder of manuscript.
Thank you very much for pointing this out. We corrected the citations style.
“Only pseudomolecular ions with a relative intensity of > 6 Í 104 in the MS signal (TIC) were subjected to subsequent MS/MS analysis.”
Comment: Please define what pseudomolecular ions are.
We added a definition in line 183 stating that all pseudomolecular ions found were [M+H]+
“Similarly, all investigated species of Lamprohiza (four species), Lampyroidea (three species), and Luciola (three species) were devoid of the compounds suggesting that the ability to produce lucibufagins …. arose before the diversification of the clade comprising the subfamilies Lampyrinae and Photurinae (Fig. 1).”
Comment: This is not the most parsimonious conclusion with respect to lucibufagin production as shown here (and using the Martin et al. 2019 phylogeny):
Thank you very much for pointing this out!! We were clearly wrong and adapted our discussion based on your suggestions (line 323-333).
“We did not detect bufadienolides in 12 specimens belonging to the family Lycidae (6 species), which are closely related to the Lampyridae [29] supporting the hypothesis that the occurrence of lucibufagins is restricted to the fireflies.”
Comment: What about Rhagophthalmidae and Phengodidae, two beetle families that are more closely related to fireflies than lycids?
This is a good point! Unfortunately, the phylogeny by Martin et al. was not yet published when our data were collected...
Table 1: Total amounts and diversity of structurally different lucibufagins
Comment: The data in this table are never presented or discussed in detail (in results or in discussion): what does the individual variation in total lucibufagins imply for using collections for detecting bioactive compounds (e.g. minimum specimen required for analysis and/or evidence of denaturation)? How confident can we be that these are functional compounds given this variation? Is this also typical for “fresh” specimen? Same for number of lucibufagin peaks? How can this inter-individual variation be interpreted (on its own and compared to total) with respect to extracting bioactive compounds (of this type) from specimens in NH collections?
At the time being, we cannot answer these questions due to our limited sampling. We could try a correlation analysis between collecting date and amount of bufadienolides but this might be highly biased since we don't know anything about the history of the beetle individuals tested... Moreover, total amounts of bufadienolides will likely depend on body mass which was not collected during our study. It seems like some of the oldest specimens also have comparatively low amounts of bufadienolides but there are also exceptions. We added this information to the discussion (line 284-289).
“Across our firefly samples, we found four pseudomolecular ions “
Comment: Please be more specific (and define terminology in methods) and explain what this is in the context of this study. Are these indicators for lucibufagins/candidates? Or false positives? It would be helpful to set the stage on how to interpret these before readers engage with the rest of the paragraph.
We are sorry for causing confusion here... We rephrased the paragraph (lines 244-252) to clarify.
Remarkably, our sample of L. brutia was collected in 1905; and we extracted lucibufagins from several specimens that also were older than 100 years (the oldest specimen was collected in 1899). Consequently, even very old museum specimens can reveal important information to address biological questions and lead to the discovery of novel natural products.
Comment: This is amazing! In terms of what we can learn from them: Is there any suggestion in total number of lucibufagins and peaks of the older specimens with respect to conservation/denaturation of lucibufagins?
Please see our explanation above and the modified discussion (line 284-289).
“Besides the four putatively novel lucibufagins, we found the following putative pseudomolecular ions ..”
Comment: Please explain (in methods and here). What is the difference between:
- a putatively novel lucibufagins and pseudomolecular ions?
- a putatively novel lucibufagins and putative pseudomolecular ions?
We added some explanation to clarify what we mean and took out "putative/putatively"
To understand the evolution of bufadienolide production in the Lampyridae it will be very informative to analyze specimens of the subfamilies Ototretinae, Psilocladinae, and Amydetinae (see Fig. 1).
Comment: Please expand reasoning (see above)
We included a phyloegentic interpretation, please see above (line 323-333).
Finally, we demonstrated that non-invasive methanol extraction of museum specimens and the analysis of the extracts using HPLC-DAD and HPLC-MS enables the discovery of new natural products, making this approach promising for large-scale screenings for natural products in natural history collections.
Comment: Please add a discussion on sources of error (see above) and how to verify products versus artifacts. Also please add how generalizable these findings on lucibufagins/cardiac glycosides are for other chemical compounds (see above).
Included, please see above.